# Performance of a 7-Type HPV mRNA Test in Triage of HPV DNA Primary Screen Positive Women Compared to Liquid-Based Cytology

Sveinung Wergeland Sørbye [1],* , Bente Marie Falang [2] and Mona Antonsen [1]

1    Department of Clinical Pathology, University Hospital of North Norway, 9019 Tromsø, Norway
2    PreTect AS, 3490 Klokkarstua, Norway
*    Correspondence: sveinung.wergeland.sorbye@unn.no; Tel.: +47-77-62-72-23

**Abstract:** Background: A plethora of data supports HPV-based screening to be the preferred strategy for cervical cancer prevention. The shift to a more sensitive first-line test brings the need of effective triage up for discussion. Currently, most algorithms apply cytology as a triage of HPV-DNA positive women. This study compared the performance of a 7-type HPV-mRNA test to cytology. Methods: From 1 January 2019 until 31 December 2021, cervical samples from 58,029 women were examined at the University Hospital of North Norway. A total of 30.5% (17,684/58,029) fulfilled the criteria for HPV-DNA primary screening. All positive samples were triaged by cytology and followed-up according to national guidelines through 2022. Additionally, a 7-type HPV-mRNA test was applied. The study endpoint was a histologically confirmed high-grade lesion (CIN2+). Results: A total of 5.6% (990/17,684) had positive HPV-DNA test, 97.2% (962/990) with valid HPV-mRNA results. A total of 55.5% (534/962) had abnormal cytology (ASC-US+), and 35.1% (338/962) had a positive HPV-mRNA test. A total of 13.9% (134/962) had CIN2+. The sensitivity (CIN2+) of cytology versus the HPV-mRNA test was 76.1% (102/134) versus 73.1% (98/134), $p = 0.67$. The specificity was 47.8% (396/828) versus 71.0% (588/624), $p < 0.001$. PPV was 19.1% (102/534) and 29.0% (98/338), $p < 0.001$, respectively. The number of colposcopies per CIN2+ detected by cytology and HPV-mRNA test was 5.2 and 3.1. Conclusion: The 7-type HPV mRNA test was significantly more specific than cervical cytology in a triage of HPV-DNA positive women. Using this biomarker as the threshold for colposcopy may better balance the benefits and harms of screening.

**Keywords:** HPV-DNA; primary screening; HPV-mRNA; biomarker; triage; liquid-based cytology; colposcopy; CIN2+

## 1. Introduction

Today's comprehensive knowledge and applied science about HPV as the causative agent for cervical cancer offers enormous potential for improved prevention of cervical cancer [1]. For decades, cytology-based screening has been the most important strategy, reducing cervical cancer incidences and mortality by identification and treatment of precancerous lesions [2]. However, the availability of new technologies and HPV-vaccination urges policymakers to review current screening policies and adapt practices. The transition to HPV primary screening is a logical step considering the overwhelming evidence from clinical data that HPV testing has higher sensitivity than cytology for cervical cancer and its precursors, implying increased safety [3]. However, its specificity for CIN2+ is lower than cytology [4].

Understanding the natural history of the HPV infection and progression to cervical cancer is key to tailoring an optimal programme for prevention, balancing the benefits and harms of screening [5]. HPV is a common infection, highly prevalent among the general population, declining with age. The risk of future cell abnormalities, linked to the overexpression of the E6/E7 viral oncogenes affecting cellular pathways, varies strongly by

HPV genotype and duration of infection [6]. Most of the infections are transient and will be eliminated by the immune system within 1 to 2 years, resulting in a considerable risk of overdiagnosis and overtreatment of the HPV primary screening positives [7].

Several countries in Europe have implemented molecular testing, yet it is still debated how to best manage the HPV-screened positives [8]. To date, cytology alone or in combination with partial genotyping (HPV 16/18) are the recommended triage options. Despite this, cervical cytology is an imperfect test due to its subjectivity and poor reproducibility where its sensitivity varies widely, reducing the overall programme sensitivity [9].

Given the above, the onus is still on the expert community to search for alternatives for how to improve management of the HPV-positive women. In a recent systematic review by Bonde et al., they examined if extended HPV genotyping might be used as risk discrimination of high-grade cervical intraepithelial neoplasia. Evidence was presented that seven HPV types (16, 18, 31, 33, 45, 52, 58) represented a significantly higher risk of CIN3+, regardless of the cytology result at baseline [10]. The genotype-specific risk profiles identified have been supported by literature describing the same seven genotypes to be the most oncogenic HPV types, identified in 90% of cervical cancer tissue worldwide [11]. Wentzensen et al. further discuss how the clinical management of women with cervical cancer screening results is moving to use risk thresholds rather than individual test results, describing the goals of screening: "An optimal integrated screening and triage strategy should reassure the vast majority of women that they are at very low risk of cervical cancer, send the women at highest risk to colposcopy at the right time, when disease can be colposcopically detected, and minimize the intermediate risk group that requires continued surveillance" [12].

One of the promising biomarkers that might serve as a better risk stratification tool among HPV primary screen positive women warranted for immediate colposcopy is the detection of HPV mRNA E6/E7 [13]. A new HPV mRNA test on the market enables extended genotyping of the seven types (16, 18, 31, 33, 45, 52, 58) proven to be associated with the highest oncogenic risk. The knowledge of the individual genotype and its risk profile, combined with the information on whether mRNA E6/E7 is overexpressed or not, might be a more precise threshold for the colposcopy of HPV primary screening positive women. In this study, we compared the performance of the 7-type HPV mRNA test to cervical cytology being the approved strategy in a triage of the HPV-DNA primary screen positive women and evaluated the test characteristics towards the detection of cervical intraepithelial neoplasia grade 2 (CIN2+).

## 2. Materials and Methods

### 2.1. The Norwegian Cervical Cancer Screening Recommendations during the Study Period

Norway has a nationwide organised screening system, the Norwegian Cervical Cancer Screening Programme (NCCSP), which is run by the Cancer Registry of Norway. All women between the ages of 25 and 69 years are invited to participate, and reminders are sent to non-attenders. Cervical specimens are sent for centralized testing at national hospitals' pathology departments, which perform HPV DNA testing and liquid-based cytology (LBC) evaluations. LBC results are reported following the 2014 Bethesda classification [14], and histology is reported according to the WHO's CIN classification [15]. Norwegian recommendations instruct biopsies to be obtained on all visible lesions; if no lesions are visible, then they will be obtained from all four quadrants of the squamocolumnar junction (SCJ).

The results of primary testing and triaging are evaluated, and in the event of any abnormal test result, follow-up actions according to the Norwegian algorithm are recommended. Recently, HPV DNA primary screening for women $\geq$ 34 years of age every five years was implemented, while primary cytology every three years has been maintained for women 25–33 years of age. Following the governmental roll-out plan for a gradual introduction of HPV-DNA test in primary screening, 50% of the women 34–69 years of age were randomized for HPV testing from 2019 to 2020, whereas HPV testing was fully

implemented from 2021 [16]. HPV-DNA primary screen positive women were managed according to a standard of care as follows; HPV 16/18 DNA-positive/cytology abnormal and HPV (non-16/18) DNA-positive/high-grade cytology-positive women were sent to colposcopy/cervical biopsies evaluations to determine disease status. Women with other HPV/cytology results combinations were sent to 1- or 2-year surveillance to determine HPV persistence [17].

### 2.2. Study Design

This study was nested in a real HPV-primary screening setting in the northernmost county in Norway from 2019 to 2021. The department of Clinical Pathology at the University Hospital of North Norway (UNN) receives cervical samples collected in ThinPrep Pap test vials with PreservCyt solution from all women participating in cervical cancer screening in Troms and Finnmark county [18]. All women 34–69 years of age who were randomised for HPV-DNA primary screening at UNN during the three-year period were included. The department utilised the Cobas 4800 HPV-test (Roche) in daily routine, which reports individual results for HPV 16 and HPV 18, along with a simultaneous, pooled result for other high-risk genotypes (31, 33, 35, 39, 45, 51, 52, 56, 58, 59, 66, 68). Following the national guidelines standard of care, all HPV-DNA primary screen positive women were triaged by cytology and partial genotyping (HPV 16/18) results, guiding further referrals to colposcopy or surveillance testing [17]. All test positives have been followed up until December 2022 for the evaluation of histologically confirmed cervical intraepithelial neoplasia grade 2 (CIN2+).

The residual LBC-samples after HPV-DNA testing and LBC processing were kept at ambient temperature and sent to an accredited HPV-processing laboratory (PreTect AS) for HPV mRNA testing within 2–6 weeks of the date of collection. Nucleic acids were isolated from 1 mL of the residual sample using PreTect X (PreTect AS, Klokkarstua, Norway) and analysed for HPV mRNA E6/E7 expression from the types 16, 18, 31, 33, 45, 52 and 58 (PreTect HPV-Proofer'7, PreTect AS, Norway) according to the manufacturer's instructions. The assay is based on real-time Nucleic Acid Sequence Based Amplification (NASBA) technology, an isothermal RNA amplification method targeting full-length E6/E7 transcripts providing qualitative results. Assay validation comprised positive and negative controls corresponding to the viral mRNA for all targets, including an intrinsic sample control confirming sample adequacy. Results were interpreted and presented by the PreTect Analysis Software.

### 2.3. Selection of Study Population

From 1 January 2019 until 31 December 2021, cervical samples from 58,029 women were sent to the department of Clinical Pathology at UNN for examination. A total of 30.5% (17,684/58,029) of the women were randomised and fulfilled the criteria for HPV-DNA primary screening. Among those, 5.6% (990/17,684) of the women tested positive for HPV DNA and were eligible for HPV mRNA testing. Residual LBC-samples with insufficient volume (<1 mL) for HPV mRNA testing and samples testing negative for the mRNA intrinsic sample control (ISC) were excluded. In total, 2.8% (28/990) of the HPV DNA-positive women were excluded; hence, the valid study population comprised 962 women.

### 2.4. Study Outcomes

The aim of this study was to compare the effectiveness of a new type of test for human papillomavirus (HPV) mRNA, PreTect HPV-Proofer'7, with liquid-based cytology (LBC) in identifying cervical intraepithelial neoplasia grade 2 (CIN2+) in women who tested positive for HPV DNA in the initial screening. The comparison was based on several measures, including sensitivity, specificity, diagnostic accuracy (AU), and positive and negative predictive values (PPV/NPV). We defined sensitivity as the proportion of CIN2+ cases detected by either the baseline cytology or baseline HPV mRNA test. Specificity and

negative predictive values (NPV) were calculated assuming that women with negative cytology results (including repeat testing after 12 months) and negative HPV DNA test results, and who showed no signs of CIN2+ during the follow-up period, did not have the disease. Test accuracy was calculated (AU = (SE + SP)/2). Further, the increasing risk of CIN2+ by the positive predictive values (% PPV) for each test combination at the specific branching points in screening (first-line test and triage) was illustrated using a colour flow risk-diagram.

Secondly, the rate of colposcopies per CIN2+ detected was evaluated for the four alternative triage strategies by applying different thresholds for referral for the two technologies (Cytology ASC-US+ versus ASC-H+ and 7-types versus 5-types HPV mRNA test). The ratio was calculated by estimating that all the test positives within each strategy were referred for a colposcopy and divided by the actual number of CIN2+ cases detected, even though the total CIN2+ prevalence was detected solely using cytology (and repeat HPV DNA+ test) as the threshold for referral, following accordance with the Norwegian guidelines.

Finally, the absolute risk of CIN2+ by HPV genotype for DNA versus mRNA detection was calculated as the percentage of CIN2+ cases among the total number of HPV genotype infections.

Statistical analysis: Data were analysed in Statistical Package for Social Sciences (SPSS) version 28.0 with Chi-square test and Chi-square test for trend with *p*-values < 0.05 as significance level.

Ethical approval: The Regional Committee for Medical and Health Research Ethics (REC North) has approved the protocol as quality assurance (REK nord 203384). Norwegian regulations exempt quality assurance studies from the written informed consent of the patients.

## 3. Results

### 3.1. Test Positivity Rates and Detection Rate of CIN2+

Of the 17,684 women with an HPV-DNA test as primary screening, 990 women (5.6%) had a positive HPV-DNA test. Included were 962 women with a positive HPV-DNA test and a valid HPV mRNA test. In a triage of HPV-DNA positive women, 55.5% (534/962) had abnormal cytology (ASC-US+) and 35.1% (338/962) had a positive HPV mRNA test (Figure 1). Of the women with a positive HPV-DNA test, 414 women had a biopsy taken during follow-ups throughout December 2022. The total detection rate of CIN2+ among the study population was 13.9% (134/962).

### 3.2. Sensitivity, Specificity, and Predictive Values for CIN2+

In total, 134 women (13.9% (134/962)), had confirmed CIN2+ during follow-up. The sensitivity of cytology versus the HPV mRNA test in triage of HPV-DNA positive women was 76.1% (102/134) versus 73.1% (98/134), *p* = 0.67. The specificity of cytology versus the HPV mRNA test was 47.8% (396/828) versus 71.0% (588/828), *p* < 0.001. The PPV for CIN2+ was 19.1% (102/534) for cytology and 29.0% (98/338) for the HPV mRNA test, *p* < 0.001. NPV was 92.5% (396/428) and 94.2% (588/624), *p* = 0.33, respectively.

Applying high-grade cytology (ASC-H+) as the cut-off instead of ASC-US+, the number of triage positives was reduced from 55.5% (534/962) to 6.9% (66/962), while the sensitivity for CIN2+ dropped from 76.1% (102/134) to 27.6% (37/134). Restricting the number of HPV-types in the HPV mRNA test from seven (16, 18, 31, 33, 45, 52 and 58) to five (16, 18, 31, 33 and 45), the number of triage positives was reduced from 35.1% (338/962) to 27.9% (268/962), *p* < 0.001, and the sensitivity for CIN2+ dropped from 73.1% (98/134) to 64.2% (86/134), *p* = 0.15, (Table 1).

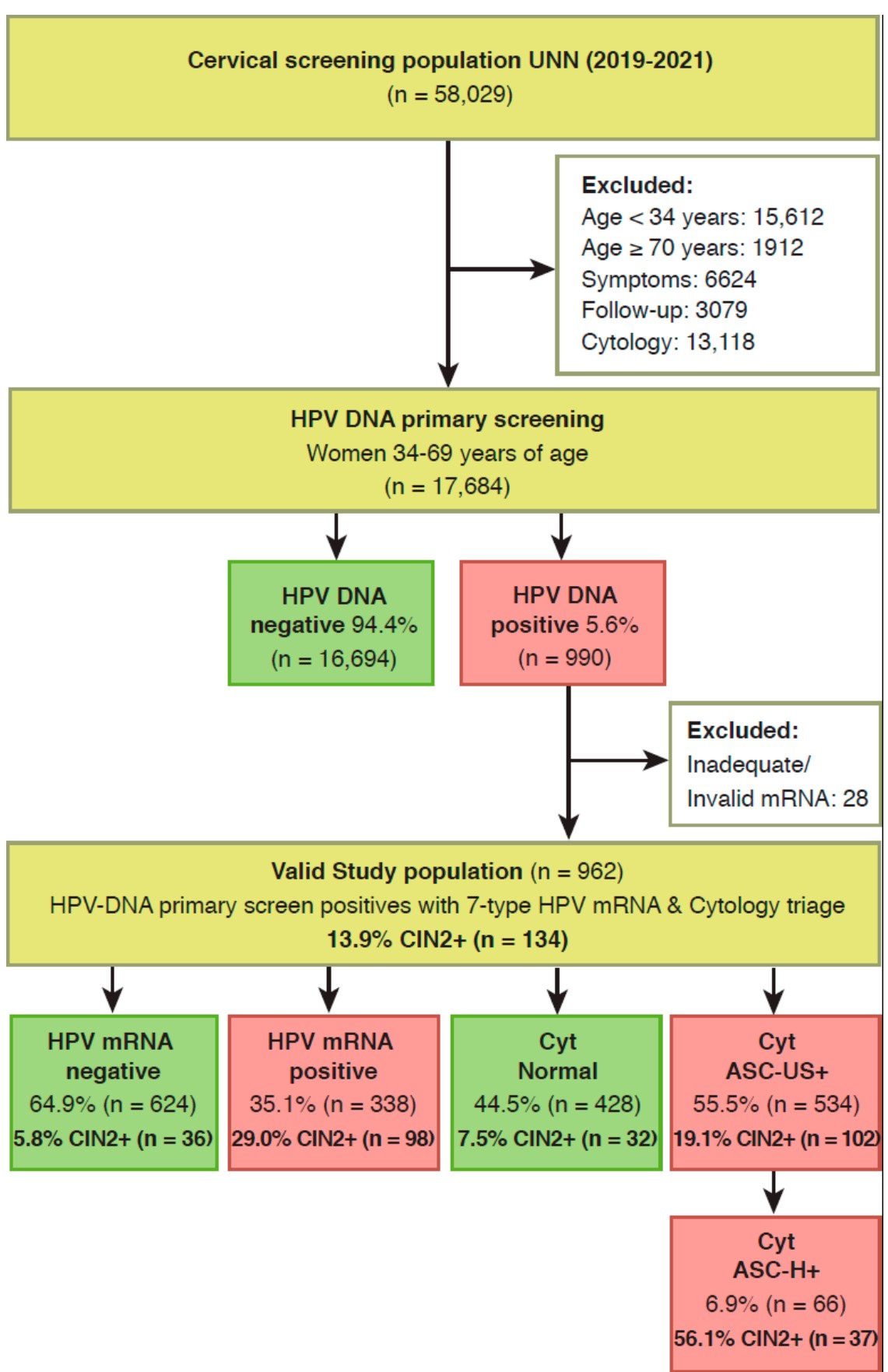

**Figure 1.** Flowchart outlining the study population and detection rate of CIN2+.

**Table 1.** Comparison of different triage strategies in hrHPV-positive women for detection of CIN2+.

| Triage Strategies | TP | TN | FP | FN | SE% | SP% | AU *% | PPV% | 95% CI (PPV) | NPV% |
|---|---|---|---|---|---|---|---|---|---|---|
| Cytology ASC-US+ | 102 | 396 | 432 | 32 | 76.1 | 47.8 | 62.0 | 19.1 | 15.8–22.4 | 92.5 |
| Cytology ASC-H+ | 37 | 799 | 29 | 97 | 27.6 | 96.5 | 62.1 | 56.1 | 44.1–68.0 | 89.2 |
| HPV mRNA 16, 18, 31, 33, 45+ | 86 | 646 | 182 | 48 | 64.2 | 78.0 | 71.1 | 32.1 | 26.5–37.7 | 93.1 |
| HPV mRNA 16, 18, 31, 33, 45, 52, 58+ | 98 | 588 | 240 | 36 | 73.1 | 71.0 | 72.1 | 29.0 | 24.2–33.8 | 94.2 |

True positive (TP), True negative (TN), False positive (FP), False negative (FN), Sensitivity (SE), Specificity (SP) and * Diagnostic Accuracy (AU) = (SE + SP)/2 for CIN2+ per triage strategy.

The increased risk of high-grade dysplasia (CIN2+) across the triage positives was 19.1% for ASC-US+ versus 29.0% for HPV mRNA-positive women. Among triage negatives, the risk of CIN2+ in cytology normal women was 7.5% versus 5.8% in HPV mRNA-negative women, as illustrated in Figure 2.

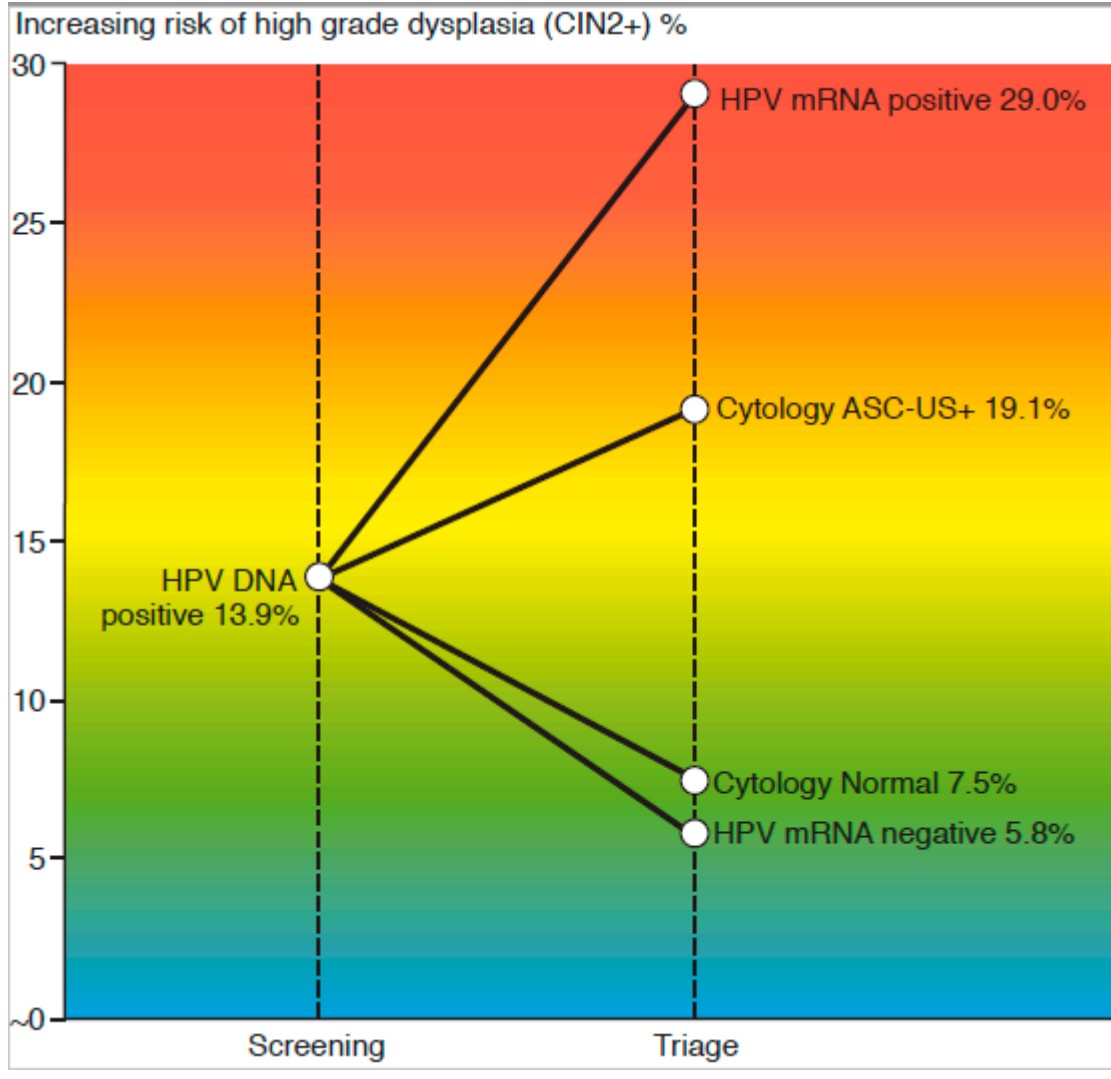

**Figure 2.** Increasing risk of CIN2+ at specific branching points in screening. The y-axis indicates CIN2+ risk (%), based on the test's PPV, and the x-axis represents the specific branching points in screening, at the time of the first screening test followed by the triage outcomes.

### 3.3. Estimated Number of Colposcopies per CIN2+ Detected

The number of colposcopies per CIN2+ detected by cytology if applying ASC-US+ or ASC-H+ as a threshold for referral was 5.2 versus 1.8, while the estimated number of

colposcopies to be performed to detect 1 case of CIN2+ based on HPV mRNA 7-type versus 5-type test positives was 3.4 versus 3.1, respectively (Table 2). The number of colposcopies required per CIN2+ case detected was calculated by estimating that all test positives for each strategy would be scheduled for colposcopy, even though the total CIN2+ prevalence among our study population was detected solely using cytology (and repeat HPV DNA+ test) as the threshold for referral.

**Table 2.** CIN2+ detection at baseline using the four triage strategies in hrHPV-positive women.

| Triage Strategies | Triage Positives (%) | No. of CIN2+ (1) | No. of Colposcopies (2) | Colposcopies/CIN2+ (3) |
|---|---|---|---|---|
| Cytology ASC-US+ | 55.5 | 102 | 534 | 5.2 |
| Cytology ASC-H+ | 6.9 | 37 | 66 | 1.8 |
| HPV mRNA 16, 18, 31, 33, 45+ | 27.9 | 86 | 268 | 3.1 |
| HPV mRNA 16, 18, 31, 33, 45, 52, 58+ | 35.1 | 98 | 338 | 3.4 |

(1) The number of CIN2+ cases detected by each strategy among the total 134 cases. (2) The estimated number of colposcopies to be performed if all test positives will be scheduled to colposcopy. (3) The calculated number of colposcopies required to detect one case of CIN2+.

### 3.4. HPV Genotype Specific Risk of CIN2+

The genotype specific risk of CIN2+ was calculated for the HPV DNA test, distinguishing between the partial genotyping that reports HPV DNA 16 and 18 with the remaining 12 types reported as a pool, and compared to the absolute risk for the 7 individual genotypes as reported by the HPV mRNA test. The estimated CIN2+ risk for women infected with HPV 16 was 34.6% detected by the DNA test versus 53.4% by the mRNA test. For HPV 18, the risks were 23.1% and 33.1%, respectively. For the pooled HPV DNA results (12 types), the CIN2+ risk was 10.1%, while for the pooled mRNA types (5 types), the CIN2+ risk was 20.9%. Among the seven individual HPV types as reported by the mRNA test, highest score was for HPV type 16 (53.4%), followed by HPV 33 (50.0%), HPV 18 (33.1%), HPV 31 (26.1%), HPV 52 (23.7%), HPV 58 (18.8%) and HPV 45 (13.6%) (Table 3).

**Table 3.** Absolute risk for CIN2+ by HPV genotype infection.

| hrHPV Genotype | No. of Infections | No. of CIN2+ | Risk Estimate (%) | 95% CI |
|---|---|---|---|---|
| 16_DNA | 130 | 45 | 34.6 | 26.4–42.8 |
| 16_mRNA | 73 | 39 | 53.4 | 42.0–64.9 |
| 18_DNA | 39 | 9 | 23.1 | 9.9–36.3 |
| 18_mRNA | 30 | 10 | 33.1 | 16.5–50.2 |
| Other_DNA * | 793 | 80 | 10.1 | 8.0–12.2 |
| Other_mRNA ** | 235 | 49 | 20.9 | 15.7–26.0 |
| 31_mRNA | 88 | 23 | 26.1 | 17.0–35.3 |
| 33_mRNA | 28 | 14 | 50.0 | 31.5–68.5 |
| 45_mRNA | 59 | 8 | 13.6 | 4.8–22.3 |
| 52_mRNA | 59 | 14 | 23.7 | 12.9–34.6 |
| 58_mRNA | 32 | 6 | 18.8 | 5.2–32.3 |

* 12 types (31, 33, 35, 39, 45, 51, 52, 56, 58, 59, 66, 68). ** 5 types (31, 33, 45, 52, 58).

## 4. Discussion

Our study is the first to analyse the performance of extended genotyping by mRNA E6/E7 detection in a triage of HPV-DNA primary screen positive women. The first objective was to compare the key performance indicators such as sensitivity, specificity, and predictive values for detection of CIN2+ towards the established triage implemented. The reported data revealed that a 7-type mRNA test significantly increased the specificity x1.5 compared to liquid-based cytology, while a similar sensitivity was maintained. The risk of CIN2+ among mRNA positives (29.0%) was substantially higher versus the ASC-US+ cases (19.1%), while the risk among triage mRNA negatives (5.8%) was lower than that of cytology (7.5%).

Molecular HPV testing has shown to be an objective, preferential method in preventing cervical cancer. While HPV DNA tests detect the presence of virus, in both a transient and persistent state, mRNA tests identify only transcriptionally active viruses. Considering that the established cause of cervical cancer is not the infection per se, but an overexpression of the E6/E7 viral oncogenes affecting cellular pathways, it might be reasonable to trust HPV mRNA tests to give more precise information of a high-risk condition, as supported by our findings. The literature describes the E6/E7 mRNA biomarkers as having a strong correlation with the risk of cervical cancer development [13,18–23]. Additionally, in contrast to cervical cytology, triage by molecular biomarkers allows the use of HPV-self-sampling [24]. Importantly, self-sampling has been anticipated to offer great potential to increase the participation of hard-to-reach women, thus improving the prevention of cervical cancer [25].

In view of HPV being highly prevalent (8–10% in western countries) and the lifetime risk for HPV estimated to be up to 80% for sexually active people combined with the biology of the viral infections in which most are self-resolving [26], the benefits of a sensitive 14-type HPV test in primary screening are challenged by the low specificity [4]. The risk of a clinically significant disease (CIN3+) among HPV-DNA positive women has been shown to be only 5–8% [27], a contrast that might result in unnecessary follow-up and overtreatment of test-positive women. Affirming the above, our study population represented a low-risk population, where all women with any previously abnormal screening results were excluded from enrolment to HPV primary screening in Norway during the implementation phase. Even so, the HPV DNA positivity rate was 5.6%, resulting in a CIN2+ prevalence at 13.9% with only 3.0% CIN3+ cases detected during follow-up.

Most countries having implemented HPV DNA test in primary screening use cytology in the triage of screen-positive women, applying ASC-US+ as the cut-off. When tailoring an optimal screening algorithm, it is important to carefully choose a threshold for referral to colposcopy based on the balance between benefits and harms. Comparing screening strategies, the number of colposcopies required to detected one CIN2+ case is commonly used as an indicator of over-referrals, representing the harms of screening. Our analysis estimated the number of colposcopies/CIN2+ across the 4 triage strategies presented, where the lowest ratio (1.8) was found for LBC using ASC-H+ as a threshold for colposcopies, increasing to 3.1 for the mRNA test, and almost tripling (5.2) for the established practice using ASC-US+ as threshold. However, these estimates must be seen in context with the fact that in this study, the total CIN2+ prevalence was solely identified by current practice using cytology ASC-US+ (and repeat HPV DNA+ test) as the threshold for colposcopies.

In countries with limited colposcopy resources, it might be justified to make use of ASC-H+ as a cut-off to reduce the number of colposcopies, whilst maintaining the ability to identify the few women warranted for immediate colposcopy and treatment. In the present material, this strategy would reduce the referral rate at baseline from 55.5% to 6.9%, at the expense of a substantial reduction of sensitivity for CIN2+ from 76.1% to 27.6% and lowering the number of colposcopies per CIN2+ from a ratio of 5.2 for ASC-US+ to 1.8 for ASC-H+. It must be noted that in developed countries, a sensitivity for CIN2+ at baseline of 27.6% is considered far too low for the implementation of such a strategy.

Ideally, optimal screening strategies would detect as many CIN3+ as possible while keeping the number of colposcopies at an acceptable level. Thus, requiring a highly specific test in the triage of the HPV screen positive women [28]. Evaluating which triage approach might perform better for HPV DNA primary screen positive women, our data confirmed the 7-type HPV mRNA test to be significantly more specific than cervical cytology (71.0% versus 47.8%), reducing the number of colposcopies while keeping the same sensitivity towards the detection of CIN2+. Our findings were in line with the previously reported advantage of mRNA testing, verifying the E6/E7 mRNA transcripts to have a higher specificity, with the potential to limit the addressed harms of screening for cervical cancer in a population of healthy women [13,18–23]. The repeated discussion of whether a slightly lower sensitivity for the mRNA-test compared to cytology (73.1% versus 76.1%, $p = 0.67$)

might negatively affect the effectiveness of screening is of minor importance as long as triage negative women are guided to surveillance testing at dedicated intervals. Realistically, no test is 100% perfect, and we must own up to the need for the periodically repeated testing of triage-negative women, informed by the tests' safety (NPV). In the present analysis, NPV for the approved triage algorithm cytology ASC-US+ was 92.5%, while the mRNA test provided similar or improved safety, (NPV 94.2%), justifying the return to follow-up for triage mRNA-negative women as well as for cytology negative women.

The recognition that a persistent HPV infection is necessary for developing precancer and cancer, and the characteristics of infections including genotype and duration of infection, underlies the new US risk-based approach for assessing a patient's immediate risk of precancer [29]. The value of HPV-extended genotyping for clinical management of abnormal screening results is well established in the literature [28]. Scientific data over decades have been informing regarding the risk profile per genotype and their cancer-causing strengths, identifying 7 hrHPV-types (HPV 16, 18, 31, 33, 45, 52, 58) as the most prevalent and aggressive types associated with cervical cancer [10,11]. These data have helped to improve the HPV vaccine by including the same seven types for maximal protection and informed the design of new molecular tests entailing extended genotyping.

Recent publications address the fact that genotyping may improve the risk stratification of hrHPV screen positive women after investigating the difference in HPV type distribution among HPV-infected women across different cervical lesions, from normal to severe lesions [30–32]. So et al. concluded that HPV 16, 31, 33, 52, and 58 infections implied significant risk of high-grade disease for cervical carcinogenesis [30], while in a Danish population-based study by Kjær et al., the greatest risk was associated with HPV 16, followed by 18, 33, 31, 35, 58 and 45 [31]. Guen et al. confirmed in their meta-analysis, reviewing HPV-infected women from cervical infection to cancer, the exceptional difference in carcinogenicity among individual genotypes. Again, the types HPV 16, 18, 31, 33 were identified as the types carrying the absolute highest risks for progression [32].

In our analysis, the HPV mRNA genotype-specific risks for CIN2+ take precedence over the corresponding HPV DNA types, verifying the importance of the seven types included in the triage test. In descending order, our data reported the highest absolute CIN2+ risk for types 16, 33, 18, 31, 52, 58, and 45, in line with the findings of Bonde et al., summarising their findings as follows: "Beyond HPV 16, 31, 18, and 33, HPV 52, 58, and 45 carried moderate risks, with 35, 39, 51, 56, 59, 66, and 68 consistently having the lowest CIN 3 or worse risks, regardless of cytology" [10].

In Europe, most cases of cervical cancer are caused by only 5 hrHPV-types (HPV 16, 18, 31, 33 and 45) [11]. Restricting the number of HPV-types in triage to mRNA expression from these five types in this study, the referral rate was reduced from 35.1% to 27.9% ($p$ < 0.001). The sensitivity for CIN2+ was reduced from 73.1% to 64.2%, ($p$ = 0.15), and the corresponding number of colposcopies per CIN2+ was 3.4 using 7 HPV mRNA-types and 3.1 using 5 HPV mRNA-types as a cut-off. Given the significant reduction in number of referrals and the insignificant drop of sensitivity, the five-type approach might be worth considering. However, the slight reduction in number of colposcopies per CIN2+ (3.4–3.1) must be weighed against the increased detection of CIN2 in a prospective cost-benefit analysis.

In the previous work of Stoler et al., various triage strategies including combinations of genotypes and dual-stained cytology were evaluated for a subset of women with hrHPV infections who were participating in the Addressing the Need for Advanced HPV Diagnostics (ATHENA) study [28]. Among the HPV-DNA genotype combinations evaluated, the 7-type (HPV16/18/31/33/45/52/58) was found to be more sensitive than the FDA-approved algorithm (HPV16/18+ or 12-other hrHPV+ AND Pap+), but also resulted in a slight increase in the number of colposcopies per CIN3 detected from a ratio of 8.4 to 8.9. Taking into consideration that the reported HPV types were detecting DNA and identified using the LINEAR ARRAY HPV Genotyping Test along with assessing CIN3+, a direct comparison to our material is not possible. However, by utilizing mRNA E6/E7-extended

genotyping, a lower relative positivity rate for the mRNA types versus the corresponding DNA is presumed, especially in a low-risk population such as women attending primary screening. In Predictors 3, Cuzick et al. reported a relative positivity rate for HPV DNA 16 at 3.5% (Cobas 4800, Roche) versus 2.1% for HPV mRNA 16 (PreTect HPV-Proofer, PreTect AS) [33]. In the present material, HPV 16 cases were 13.5% (130/962) versus 7.6% (73/962) for mRNA 16 positives. Integrating a 7-type mRNA test in triage would most likely improve the clinical utility over what has been reported for DNA-extended genotyping, having a higher specificity, and lowering the number of colposcopies per detected CIN2+. In 2015, Norway ran a pilot using Cobas 4800 HPV DNA test in primary screening in four counties [34]. All HPV DNA-positive women with abnormal cytology (ASC-US+) were referred to colposcopy and biopsy, giving a three-times higher referral rate for colposcopy at baseline in the HPV-arm compared to the cytology arm. Experiences from the pilot demonstrated that partial HPV 16/18 genotyping provided further risk stratification and reduced the over-referral of all women who had an HPV-positive and low-grade cytology (ASC-US/LSIL) result regarding colposcopy and biopsy. Women with low-grade cytology and HPV 16/18 should be referred to colposcopy, while women with low-grade cytology and other HPV types should be followed-up with a new HPV-test after 12 months [17]. From 2021, this approach was fully implemented in the Norwegian algorithm, expected to not only inevitably reduce the referral rate of HPV DNA-positive women, but also reduce the sensitivity of CIN2+ at baseline. Given the high inter-laboratory differences for cytology results observed in Norway [9,17], an objective biomarker as a 7-type HPV mRNA test might serve as a better alternative to reach the goal of screening, thereby maximizing the number of immediate CIN3+ detections and minimizing the number of colposcopies. Still, prospective studies addressing the associated costs of shifting from an already established practice such as cytology to a molecular triage test are required.

*Strengths and Limitations*

One of the strengths of this study is the continued enrolment of all women attending screening in one county in Norway, still ongoing. All results have been derived from the original liquid-based cytology sample, making a direct comparison of the triage alternatives possible. A major limitation is that follow up of test-positive women has been according to cytology (and repeat HPV-DNA test) only, complying with the approved Norwegian algorithm. Cytology-negative, 7-type mRNA-positive women were not scheduled to colposcopy/biopsy, and the reported performance of the mRNA test in this analysis might be underestimated accordingly. It should be acknowledged that in real-world clinical practice, adherence to screening guidelines is not always consistent. Unlike a controlled research study, individual patient evaluations are often performed, and the clinical history of the patient is taken into consideration in determining the need for referral to treatment. Another limitation of this study is the relatively low number of CIN2+ cases, which may increase the risk of a type II error and limit the ability to accurately compare the sensitivity of cervical cytology and the 7-type HPV mRNA test in detecting high-grade cervical lesions. It should be noted that present data do not include an HPV-vaccinated population; hence, the performance might be directly impacted by vaccinated women entering screening in 2023. Inevitably, debates on how to adapt screening practices in the future must recognise the importance of triage alternatives addressing a high-risk condition among a low-prevalence disease population.

## 5. Conclusions

In summary, this analysis found the 7-type HPV mRNA test to be significantly more specific than cervical cytology in triage of the HPV DNA-positive women. Using this biomarker as a threshold for referral to colposcopy may better balance the benefits and harms of screening. Women with a positive HPV DNA test and a negative HPV mRNA test in triage have equal risk of CIN2+ as the current approved triage practice and can safely be followed up with repeat HPV DNA testing after 12 months.

**Author Contributions:** Conceptualization, S.W.S. and B.M.F.; methodology, S.W.S. and B.M.F.; formal analysis, S.W.S.; investigation, S.W.S., M.A. and B.M.F.; resources, S.W.S., M.A. and B.M.F.; data curation, S.W.S.; writing—original draft preparation, S.W.S. and B.M.F.; writing—review and editing, S.W.S., M.A. and B.M.F.; visualization, S.W.S. and B.M.F.; supervision, S.W.S.; project administration, S.W.S. and B.M.F.; funding acquisition, S.W.S. and B.M.F. All authors have read and agreed to the published version of the manuscript.

**Funding:** This research received no external funding. HPV mRNA test kits were provided FOC by PreTect AS.

**Institutional Review Board Statement:** The study was conducted in accordance with the Declaration of Helsinki and approved by the Regional Committee for Medical and Health Research Ethics (REC North, 203384, 18 December 2020) for studies involving humans.

**Informed Consent Statement:** Patient consent was waived due to Norwegian regulations which exempt quality assurance studies from written informed consent.

**Data Availability Statement:** All data presented are available upon request.

**Acknowledgments:** The authors would like to extend our gratitude to the staff at the department of Clinical Pathology at the University Hospital of North Norway and to all the laboratory staff performing HPV DNA/mRNA testing, cytology and histopathology evaluation for their great work and collaboration during this study.

**Conflicts of Interest:** S.W.S. and M.A. declare no conflict of interest. B.M.F. is an employee of PreTect AS.

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
