# Peer review of "Performance of a 7-Type HPV mRNA Test in Triage of HPV DNA Primary Screen Positive Women Compared to Liquid-Based Cytology"

_jmp, doi:10.3390/jmp4020008_

Round 1

Reviewer 1 Report

This study follows the current necessity of finding an accurate and reproducible triage strategy for HPV+ women in cervical cancer screening, and as the main advantage, it is nested in the real screening setting of Norway. However, from my point of view, the study does not have enough power to compare the sensitivity between cytology and 7-type HPV-mRNA and does not have an appropriate study design to compare the rate of colposcopies for CIN2+detected. The sensitivity of the 7-type HPV-mRNA test was lower than the sensitivity of the cytology (76.6% versus 83.0%, p=0.36), and even though the difference is not statistically significant, this is mainly due to the small sample size (n=94). My suggestion is that the authors formulate a hypothesis and do an a-priori power analysis, and then re-do their analysis with more samples. As far as I’m concerned, although increasing the specificity is the main goal for a triage test, we should not expect a lower sensitivity than cytology because this would compromise the total performance of the screening program. Besides, all referrals to colposcopy were based on cytology, thus comparing the rate of colposcopies per CIN2+ detected seems not fair to cytology since it does not consider the number of colposcopies and the number of CIN2+ that would have been done and detected, respectively, if the 7-type HPV-mRNA test were used for colposcopy referrals. In my view, this study design is not suitable for reporting this outcome. Since the authors are trying to give insight into a reduction of colposcopy referrals per CIN2+ detected using the 7-type HPV-mRNA test, I suggest a study design in which all women positive in either the 7-type HPV-mRNA test or cytology received colposcopy. As a minor suggestion, the authors should keep consistency throughout the paper to improve comprehension of the study.

The complete report, including the specific comments, is attached.

Reviewer 2 Report

I have read with great interest this paper. Surely it can contribute to literature on the use of HPV DNA 2 primary screen positive women compared to liquid-based cytology.

I have appreciated that you have enrolled also patients with ASC-H.

The methods and results section is clear.

I have only a few concerns:

- Minor English revision is needed, especially because in some parts of the main text you use present tense in other one past tense.

- You should add diagnostic accuracy in your results section and in table.

Author Response

Reviewer 2: 

I have read with great interest this paper. Surely it can contribute to literature on the use of HPV DNA 2 primary screen positive women compared to liquid-based cytology. 

I have appreciated that you have enrolled also patients with ASC-H. 

The methods and results section is clear. 

Our response: Thank you for the encouraging comments to the value of our paper. 

I have only a few concerns: 

  1. Minor English revision is needed, especially because in some parts of the main text you use present tense in other one past tense. 

Our response: Thank you for bringing this to our attention. We will make sure that the revised manuscript is in line with correct English grammar and checked by native English-speaking co-workers.  

  1. You should add diagnostic accuracy in your results section and in table. 

Our response: We completely agree and have added test accuracy calculated by the number of true positives, false positives, false negatives and true negatives and stated AU= (Sensitivity + Specificity)/2 within Table 1 (L 249-251), as informed about in the study outcome section (L 177-187). 

Round 2

Reviewer 1 Report

Even though my comments and suggestions were addressed, the main limitation persists. There is a big chance of a type II error in this analysis. The authors should at least include the low number of CIN2+ as a real limitation to compare sensitivity

Author Response

Reviewer 1

Even though my comments and suggestions were addressed, the main limitation persists. There is a big chance of a type II error in this analysis. The authors should at least include the low number of CIN2+ as a real limitation to compare sensitivity

Our reponse: Thank you for all your efforts thoroughly reviewing this manuscript and suggesting improvements on how to better present our work. We are happy that you find the revised manuscript to address your initial comments and suggestions. We have now included the chance of type II errors to the subsection “Strenghts and limitations” (L 402-405).

Reviewer 2 Report

All changes have been made

Author Response

Reviewer 2

All changes have been made

Our response: Thank you for taking the time to review this manuscript, we are pleased that you find our revisions to be sufficient.